# Eco-UHPC as Repair Material—Bond Strength, Interfacial Transition Zone and Effects of Formwork Type

**DOI:** 10.3390/ma13245778

**Published:** 2020-12-17

**Authors:** Ankit Kothari, Magdalena Rajczakowska, Thanyarat Buasiri, Karin Habermehl-Cwirzen, Andrzej Cwirzen

**Affiliations:** Building Materials, Department of Civil, Environmental and Natural Resources Engineering, Luleå University of Technology, 97187 Luleå, Sweden; magdalena.rajczakowska@ltu.se (M.R.); thanyarat.buasiri@ltu.se (T.B.); karin.habermehl-cwirzen@ltu.se (K.H.-C.); andrzej.cwirzen@ltu.se (A.C.)

**Keywords:** Eco-UHPC, NSC (normal strength concrete), UPV, bond test (pull-off), SEM-EDS, ITZ, formwork, self-healing

## Abstract

A reduced carbon footprint and longer service life of structures are major aspects of circular economy with respect to civil engineering. The aim of the research was to evaluate the interfacial bond properties between a deteriorated normal strength concrete structure and a thin overlay made of Eco-UHPC containing 50 wt% of limestone filler. Two types of formwork were used: untreated rough plywood and surface treated shuttering plywood. The normal strength concrete elements were surface scaled using water jets to obtain some degradation prior to casting of the UHPC overlay. Ultrasonic pulse velocity (UPV), bond test (pull-off test), and Scanning Electron Microscopy (SEM) combined with Energy Dispersive Spectrometry (EDS) were used for analysis. Elements repaired with the Eco-UHPC showed significantly improved mechanical properties compared to the non-deteriorated NSC sample which was used as a reference. The bond strength varied between 2 and 2.7 MPa regardless of the used formwork. The interfacial transition zone was very narrow with only slightly increased porosity. The untreated plywood, having a rough and water-absorbing surface, created a surface friction-based restraint which limited microcracking due to autogenous shrinkage. Shuttering plywood with a smooth surface enabled the development of higher tensile stress on the UHPC surface, which led to a more intensive autogenous shrinkage cracking. None of the formed microcracks penetrated through the entire thickness of the overlay and some were partly self-healed when a simple water treatment was applied. The project results showed that application of UHPC as repair material for concrete structures could elongate the lifespan and thus enhance the sustainability.

## 1. Introduction

Strategic infrastructure built of concrete including road bridges, piers, harbor, and offshore structures often show durability problems when exposed to harsh environments. Deterioration of concrete is commonly related to its porous microstructure and deleterious chemical change in the binder matrix. One method to elongate the lifespan of an affected structure is to cover it with an external protective layer made of a more durable material. Ultra-High Performance Concrete (UHPC) is one of the most promising materials for such applications [1]. Its high compressive strength (>150 MPa), tensile strength (>8 MPa) combined with enhanced flowability, and very dense microstructure are the key characteristics [2,3,4,5,6,7,8,9]. However, an application of UHPC as an overlay on an old deteriorated concrete raises the question about the strength and long-term effects of the interfacial transition zone between new and old concrete (ITZ). The properties of the formed ITZ depend on several factors including e.g., substrate roughness, moisture content, and *w/c* ratio [10,11,12,13]. To mimic the surface deterioration of old-normal strength concrete, jack hammering, sand blasting, and water jetting are the commonly adopted methods [3,13,14]. However, the intense microcrack formation on the substrate from jack hammering showed relatively lower bond strength development compared to sand blasting and water jetting methods [13]. Zhang et al. evaluated the interfacial bond properties between UHPC and normal strength concrete (NSC) having various moisture degrees and substrate roughnesses using slant shear, split tensile, and direct tensile test. Results showed that rough and adequately wetted surfaces of the substrate concrete lead to a much higher bond strength when compared to a smooth substrate [7].

Curing is yet another factor, which directly influences the bond properties. UHPC covered NSC which was cured at high temperature produced high early shrinkage and thus decreased the interfacial bond compared to ambient temperature cured specimens. Furthermore, interfacial shear properties of Z-shaped single sided push-out test were determined on untreated, rough, and water jet grooved substrate specimens. The bond strength was higher for rough and grooved substrates [15,16]. Other studies indicated that a 25–50 mm thick overlay is the optimum considering material consumption, dead load, and mechanical properties [17,18,19]. Thicker overlays are highly susceptible to early age shrinkage cracks [20]. A traditional UHPC has a very high cement content (900–1000 kg/m^3^) compared to ordinary concrete (320–450 kg/m^3^). This cement consumption consequently emit 2–3 times more CO_2_ and thus significantly higher carbon footprint [21,22]. It has a very low w/b and thus a limited hydration degree of 40–50% is achieved. The unreacted Portland cement acts only as a filler [23,24,25]. Hence, its partial replacement with, for example, limestone filler will not only reduce the CO_2_ emission, but will also lower the production cost, lower the shrinkage, and reduce the crack risk [26,27,28,29,30,31]. The limestone filler replacing Portland cement in UHPC tends to affect its fresh and hardened state properties. It accelerates the cement hydration by the nucleation effect and subsequently fills the voids [32,33,34]. Calcium carbonate was observed to react with C_3_A to form calcium carboaluminate (CCA), which tended to increase the compressive strength [29,35,36,37].

The primary aim of this research was to study the interfacial bond behavior between treated normal strength concrete (NSC) and repair ECO-UHPC overlay and to determine the effects of the used type of formwork.

## 2. Experimental Setup

### 2.1. Materials and Specimen Preparation

Normal strength self-compacting concrete (NSC) was used to produce the reinforced specimens. NSC was produced by locally available ready-mix concrete distributor (Snells Betong och Ballast, Luleå, Sweden). Portland cement type CEM II/A-V 52.5N was produced by “Cementa” (Skövde, Sweden), as superplasticizer MasterGlenium SKY 823 (BASF AB, Rosersberg, Germany) was used. The maximum aggregate size was 16 mm. The Eco-UHPC concrete contained Portland cement CEM I 42.5N, condensed micro-silica 920D from Elkem (Oslo, Norway), limestone powder “Nordkalk Limus 40” from Nordkalk AB, Norquartz 45 from Sibelco Nordic (Lillesand, Norway) as well as micro sand B15 (150 µm) and B35 (350 µm) provided by Baskarpsand AB (Habo, Sweden). A polycarboxylate-based superplasticizer type “MasterGlenium ACE 30” from BASF (Rosersberg, Germany) was added. Steel fibers having lengths of 6 mm and 13 mm were provided by Krampe Harex-Germany (Hamm, Germany). The detailed mix proportions and mechanical properties are shown in Table 1. The chemical composition of the dry materials and properties of steel fibers are shown in Table 2 and Table 3.

Test samples included twelve cubes having dimensions of 150 × 150 × 150 mm^3^ and three steel-reinforced concrete columns of dimension 300 × 300 × 2500 mm^3^ (Figure 1a and Figure 2a). Four reinforcing steel bars having a diameter of 8 mm with suitable concrete cover were placed in corners of the columns. The NSC specimens were stored in laboratory conditions for a period of three months. One month before casting the UHPC overlay, the external surfaces were scaled by water jetting (Figure 1b and Figure 2b). The overlay thicknesses of UHPC were 1.5 cm and 3 cm, respectively, for cubes and columns. All sides excluding top and bottom were covered.

Two types of formwork—untreated plywood (UTT) (rough surface) and shuttering plywood (STT) (smooth surface)—were used for casting composite columns (Figure 3). The UHPC layer was casted by pouring from the top of the column. No external vibration was used. All samples were covered with a plastic foil for 24 h, followed by surface curing using water sprinklers for seven days. No special technology for laying UHPC or dismantling the formwork was used. Normal construction methods widely used in the construction field for casting or dismantling formwork were adopted.

UHPC was dry mixed for 2 min, followed by an addition of water with superplasticizer and mixed for another 5–6 min. Steel fibers were added and mixed for another 2 min. In total, the mixing time was approximately 9–10 min.

### 2.2. Testing Methodology

Fresh properties of Eco-UHPC were evaluated using the slump flow test, in accordance with the SS-EN 12350-8:2019 standard [38]. The compressive strength was measured on 100 × 100 × 100 mm^3^ cubes at 1, 7, and 28 days. The SS-EN 12390-3:2019 standard was followed, and the applied loading rate was 10 kN/s [39]. The compressive strength of composite cubes having dimensions of 180 × 180 × 150 mm^3^ was measured at 7 and 28 days after casting of the UHPC layer. The reference NSC cubes of 150 × 150 × 150 mm^3^ were tested at the same day as the composite concrete samples. The substrate concrete and the NSC-only sample were of the same age. For larger cube samples, the loading rate was set to 13.5 kN/s, following the SS-EN 12390-3:2019 standard [39]. The flexural strength of the composite and NSC specimens were determined 28 days after casting the Eco-UHPC layer using a three-point bending setup following the SS-EN 12390-5:2009 standard [40]. The used displacement rate was 0.03 mm/s. The distance between supports was 1500 mm. The cross section of test specimens was d_1_ = d_2_ = 300 mm and d_1_ = d_2_ = 360 mm for NSC and composite concrete, respectively.

The ultrasonic pulse velocity (UPV) test was used to detect internal cracking and debonding between layers of new and old concrete following the SS-EN 12504-4:2004 standard [41]. The UPV test was performed on composite columns after flexural tests and before the bond strength test, using a Pundit Lab ultrasonic instrument (Proceq, Zurich, Switzerland) with exponential transducers at 54 kHz. The direct transmission method was used to analyze the transit time on nodes (Red A-A^1^, Blue B-B^1^) as described in Figure 4a, having a path length of 36 cm. Three consecutive readings were noted on each node length without lifting the transducers and for every 10 cm from the center (0 cm) up to 100 cm in both directions as shown in Figure 4a. Furthermore, the noted readings were divided into top, mid, and bottom sections as per column level. It was assumed that the mid-section of the test column, about 40 cm in length, was damaged and expected to have lower pulse velocity or higher transit time. This mid-section corresponds to the loaded zone in the flexural test and is marked in Figure 4a, as the black part of the column.

The pull-off test was performed to measure the interfacial bond strength between the substrate concrete and the repair UHPC overlay. This test is a conventional way to analyze the bond strength, as it neglects the friction between interfacial surfaces or any other form of stresses [7,42]. The bond strengths of the two composite column specimens cast against different formworks were analyzed according to the standard ASTM C1583 [43]. A commercially available “Proceq dy-216” instrument (Proceq, Zurich, Switzerland) with 50 mm aluminum disc at a loading rate of 35 kPa/s was used to perform the test. In addition, 60 mm deep cores, including the substrate depth of 30 mm, having a diameter of 50 mm were drilled into the columns. In total, 18 tests were performed per column—three at the top and three at the bottom sections, following three sides (A, B, and C) as shown in Figure 4b,c. The bond strength was calculated using Equation (1):(1)$$f_{P}=\frac{4F}{\pi d^{2}}\;\;\;\;MPa$$
where *f_P_* = bond (pull-off) strength (MPa); *F* = failure load (N); *d* = diameter of core (mm).

Bond failure modes were classified into four types determined by a visual observation, Table 4. This classification was later confirmed by image processing techniques on broken core samples. The FIJI ImageJ software (Version 1.53C) was used for this analysis [44]. The broken core areas are either covered completely with NSC fragments or partially with UHPC-NSC fragments or only with UHPC fragments. The area fraction or percentage of NSC fragments attached to the UHPC layer was estimated using ImageJ software. The calculation was performed on the binarized image of the core surface, showing UHPC as black pixels and NSC as white pixels. A sample was categorized as substrate failure when the interface zone with the fraction of NSC attached on UHPC ranged between 85–100%. 0% NSC with smooth bond surface and no fracture in either layers were considered as interface failure. NSC area fraction ranging between 1–85% was considered as a partial interface–substrate failure. A failure in the overlay layer was classified as an overlay failure.

The crack depth in the UHPC overlay was assessed for the composite columns. A layer of epoxy resin was applied on randomly selected cracks (Figure 5a,b). After the resin had cured, 35 mm diameter cores were extracted reaching also the substrate concrete. (Figure 5c). The crack depth was determined using a digital optical microscope type Dino-Lite Pro AM-413T (Dino-Lite Europe, Naarden, The Netherlands).

Note: The diameter of core used for pull-off test was 50 mm, as per ASTM C1583 standard—while, to measure the depth of shrinkage cracks penetration, separate cores of 35 mm were drilled, followed by visualizing the macrocracks using digital optical microscope Dino-Lite Pro AM-413T.

The interfacial transition zone between NSC and UHPC was analyzed using a Scanning Electron Microscope (SEM)—JSM-IT100 (JEOL Ltd., Tokyo, Japan) coupled with an energy-dispersive spectrometer (EDS) from Bruker (Bruker Corporation, Billerica, MA, USA). For the analysis, a piece of the interface extracted from the core was immersed in isopropanol for 24 h to stop the hydration, followed by 12 h of drying. The specimens were vacuum-impregnated with a low-viscosity resin. After hardening of the resin, the specimens were polished using a set of grinding plates and diamond spray with particles sizes of 9, 3, and 1 µm with lamp oil as a lubricant and coolant [45]. SEM images were obtained in the backscattered electron mode at 150×, 2000×, and 5000× magnifications. At 150×, five images were taken on either side of the interface, followed by two consecutive layers accounting for an overall interface line length of 2700 µm and interfacial zone width of 1600 µm. The obtained images were connected and processed together using the ImageJ software [44,46,47]. Similarly, at 2000× magnification, two consecutive rows composed of 17 images/row covering the area next to the bond line were taken. The total length of the analyzed interface line was approximately 720 µm. The 120 µm wide studied interfacial zone was extending from the interface line into the fresh cast concrete. The obtained images were connected, and the interfacial zone was divided into 10 µm wide sections. The aim was to evaluate variations in the phase composition with increasing distance from the interface [48,49]. Unhydrated cement and porosity were segmented from images based on the greyscale histogram and quantitatively evaluated using the ImageJ software. In addition, SEM-EDS at 5000× magnification generated information about the chemical composition. Variations of the Ca/Si atomic ratio with increasing distance from the interface were determined to identify the chemical composition of the calcium silicate hydrate (C-S-H). The analysis was performed in the ITZ and in the bulk binder matrix. The ITZ was divided into five zones each having a width of 25 µm. The bulk binder was located approximately 6 mm from the interface. Ten areas each containing around 20 measurement points were evaluated for each zone. Measurements points were chosen based on the grey scale corresponding to the C-S-H [45]. The following settings were used for the SEM analysis: accelerating voltage of 15 kV, beam current of 50 mA, vacuum of 30 Pa, and the counting time for the SEM-EDS data points was set to 50,000 counts per analysis [45]. The collected data were limited to points with a Ca/Si atomic ratio between 0.8–2.0 [45,50]. Since the substrate concrete is comprised of exposed coarse aggregate and binder, it accounts for two different interface mechanisms at surfaces attached with UHPC. Exposed old aggregate-UHPC interface and old binder-UHPC interface were investigated.

Self-healing of shrinkage-induced cracks was observed on the surface of the UHPC layer, which was subjected to water curing. The images of the healed cracks were acquired approximately 4 months after curing. A digital light microscope, type Dino-Lite Pro AM-413T with 1.3 MP camera and a field of view of 1280 × 1024 pixels was used to obtain the crack photographs.

## 3. Results and Discussion

### 3.1. Fresh and Hardened Properties

The slump flow of UHPC mix reached 800 mm for concrete without fibers and 900 mm when fibers were present. Steel fibers presumably enhanced the breaking and dispersion of the condensed silica fume during the mixing process. The small size of the fibers and the small amount prevented the occurrence of the interlocking mechanism commonly present for regular steel fibers [51,52]. The compressive strength results of the UHPC mixes with and without steel fibers are shown in Figure 6. Specimen with steel fibers developed the maximum strength of 154.25 MPa after 28 days, which was approximately a 28% increase in comparison with samples without fibers.

The comparison of compressive strength values measured for NSC and composite concrete cubes (NSC with UPHC overlay applied on 4 sides) is shown in Figure 7a. The composite specimens developed a maximum strength of 67.93 MPa and 76.33 MPa after 7 and 28 days of curing, respectively, which was higher than for the NSC. However, the total cross section of the test composite specimens was increased which hinders a direct comparison of the results. The flexural strength of the composite specimen using a three-point bending test reached a strength of 7.96 MPa after 28 days from the application of UHPC overlay. The strength developed was 66% higher than for the NSC specimen—crushed on the same day as the composite. The NSC specimen and substrate were approximately four months old during the testing date. Load vs. displacement for composite and NSC specimens is shown in Figure 8a. Both samples showed a similar cracking behavior, Figure 8b. The ultimate measured flexural loads increased from 57.6 kN measured for the reference specimen to 165 kN for the composite repaired specimen. No delamination or debonding of the UHPC overlay was observed in any broken specimens, Figure 7b. The result can be related to the high bond strength [15,53,54,55,56,57]. It can be assumed also that the ITZ could possibly undergo further densification due to a low hydration degree and ongoing pozzolanic reactions of silica fume leading to an improved bond strength [7,58,59,60,61]. In addition, the used limestone powder could promote formation of carboaluminate and C-S-H, [62].

### 3.2. Ultrasonic Pulse Velocity (UPV)

The average measured transit time from composite columns is shown in Figure 9. In general, the transit time tended to increase with the height of the measured point over the bottom of the column. Column cast in shuttering plywood showed an average of 3.7% higher pulse velocity than when the untreated plywood was used. Additionally, pulse velocity was in the range between the solid-intact UHPC (4900 m/s) and NSC (4740 m/s) for column using shuttering plywood. Unlikely, the untreated plywood used column showed lower pulse velocity, less than the solid-intact NSC (4740 m/s). Presumably, the untreated plywood absorbed a part of the mixing water from the fresh UHPC in adjacent layers. This resulted in a lower water to cement ratio, lower hydration degree, and thus higher porosity. A higher porosity is known to slow down the transit time [63].

### 3.3. Bond Strength (Pull-Off Test)

The bond strength between NSC and the UHPC overlay tested on different levels of the casted columns varied between 2.0 and 2.7 MPa (Figure 10). The obtained bond strength results falls under the excellent bond quality as per ACI 546 (Figure 10) [64,65]. The failure occurred predominantly on the weaker NSC marked as “SBF” in Table 5. Some of the pulled-off samples showed also partial failure on interface between the aggregates originating from NSC and exposed by water jetting and the UHPC overlay, Figure 11b. The effects of the used formwork plywood were negligible in all cases. These results further confirm that the higher water absorption observed in the untreated plywood formwork increased the porosity only on the surface and did not affect the measured bond strength. The tensile strength of the overlay material was significantly higher than the developed ultimate tensile stress [66].

The bond strength is mostly affected by the substrate roughness and moisture content. According to Benoit Bissonnette et al., the dry substrate concrete tends to absorb water from the fresh overlay concrete leading to a lower hydration degree and weaker bond strength [54,67]. Others showed that moderate moisturizing of the substrate has negligible impact on bond strength. On the contrary, saturating the substrate surface could result in an excessive *w/c* ratio directly on the interface. This could lead to a lower bond strength [66,68].

The UHPC overlay showed autogenous shrinkage cracks. Interestingly, the number of cracks was lower on surfaces casted against the untreated plywood (UTT). The formed surfaces were also visibly rougher in comparison with elements casted against the smooth shuttering plywood. None of the studied cracks penetrated the full depth of the UHPC overlay, Figure 12. These results indicate that a rough surface of the untreated plywood created additional surface restrain, which anchored external layers of the UHPC and lowered the maximum developed tensile stresses. This limited the crack formation due to the autogenous shrinkage. Later, when the formwork was removed, the tensile strength of the binder matrix was sufficiently high to prevent microcracking.

### 3.4. Scanning Electron Microscope (SEM)

SEM analysis of the polished and resin-impregnated samples did not detect significant differences between samples cast against the STT and UTT plywood formwork, Figure 13 and Figure 14. The porosity was only slightly increased at the interface in both cases with visible few entrapped air voids located next to the interface as well in the bulk binder zone. Image analysis enabled approximating the fluctuation in porosity and the amount of unhydrated cement particles with increasing distance from the interface, Figure 15 and Figure 16. The air voids and cracks resulting from the specimen preparation were excluded from the analysis. The porosity at the interface was approximately 6% and tended to decrease to around 2% in the bulk binder paste regardless of the used formwork, Figure 15. In both cases, the amount of unhydrated cement particles tended to be lower close to the interface, which is related to the wall effect and to a higher local water to cement ratio, Figure 16 [69,70].

The chemical composition of the C-S-H was analyzed using SEM-EDS as a function of the distance from the interface. The analyzed spots were chosen based on the grey level in the SEM-BSE images corresponding to C-S-H only. Both specimens (column using untreated and shuttering plywood) showed a slightly higher Ca/Si atomic ratio at the binder-UHPC interface compared to the aggregate–UHPC interface, Figure 17. This is attributed to the wall effect. The aggregate–UHPC interface zone accounting locally high *W/C* and pores filled with the concentration of Ca(OH)_2_ reacted with silica fume to produce silica rich C-S-H. On the contrary, higher water absorption occurred at the NSC binder-UHPC interface resulted in the locally lower *W/C* ratio and partially inhibited the reaction with silica fume—thus, relatively showing higher calcium content in C-S-H corresponding to binder–UHPC interface. The Ca/Si ratio varied between 1.2 and 1.6. In general, these results showed that the formed ITZ is very much narrow and does not exceed 20 μm, which resulted in a very good bond strength and failure was only observed on the weaker NSC substrate side.

### 3.5. Self-Healing Efficiency

Several cracks were found to be partially filled with light-colored healing products, Figure 18b–f. Only cracks with an average width smaller than 50 µm were fully healed, Figure 18a. The healing products morphology presumably corresponded to the calcium carbonate crystals as explained in [71]. The limited healing of the UHPC can be contributed to several factors, e.g., dense microstructure of the UHPC preventing ions transport to the crack [71] or application of continuous water exposure (e.g., Qian et al.) [72]. However, due to a high amount of unhydrated cement and limestone particles, the material has self-healing potential, which could be further improved by the application of a more efficient curing regime. Recent results have shown that curing of cracked concrete with a mixture of water and retarding admixture can lead to a successful self-healing [73].

## 4. Conclusions

The aim of this study was to characterize the interfacial transition zone forming between water jetted NSC concrete and UHPC overlay simulating repair strengthening operation of deteriorated concrete columns. Additionally, the effects of formwork plywood to limit the development of autogenous microcracks were determined. The following conclusions were drawn:No delamination between the layers, despite loading parallel to the bond surface being observed indicating the bond strength exceeding the strength of the weaker NSC.Bond strength varied between 2 and 2.7 MPa regardless of the used formworkThe type of the plywood used as formwork affected the microstructure and crack formation of the external layers of the UHPC overlay.The rough and water-absorbing untreated plywood reduced the *W/C* ratio on top of the UHPC surface layer leading locally to a lower hydration degree. At the same time, the surface roughness created a surface friction-based restraint which limited the microcracking due to the autogenous shrinkage.Shuttering plywood with smooth surface enabled the development of higher tensile stress on the UHPC surface, which led to more intensive autogenous shrinkage cracking.None of the autogenous shrinkage related microcracks developed on the UHPC surface penetrated through the entire thickness of the UHPC overlay.Detected ITZ was very narrow, less than 20 μm. The estimated Ca/Si based on the SEM-EDS analysis and corresponding to the C-S-H was uniformly distributed on the interface as well as in the bulk zone with only minor statistically insignificant variations.The UHPC overlay showed signs of a limited self-healing when treated with water.Repair of degraded NSC using UHPC overlay has a high potential for full-scale application, due to its enhanced bond strength, dense microstructure.

## Figures and Tables

**Figure 1 materials-13-05778-f001:**
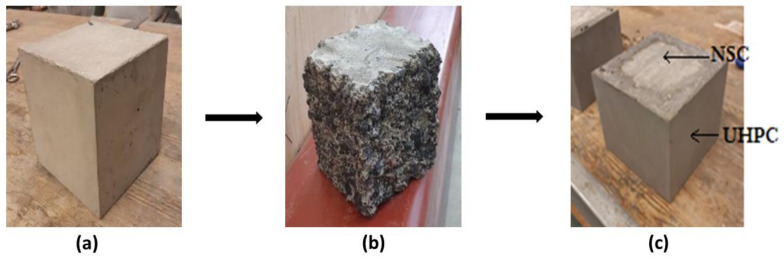
(**a**) NSC cubes; (**b**) water jet scaled cubes; (**c**) repair/rehabilitated cubes.

**Figure 2 materials-13-05778-f002:**
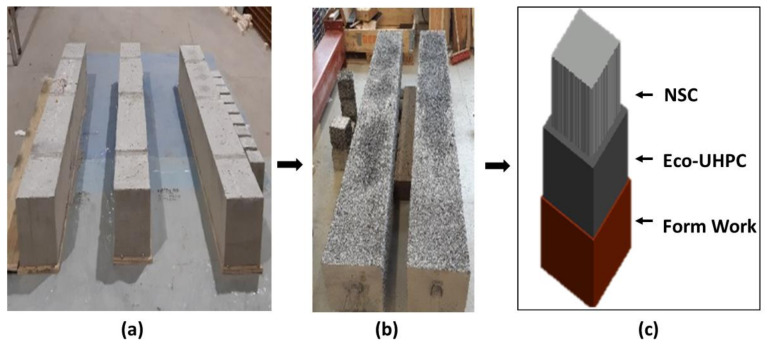
(**a**) NSC columns; (**b**) water jet scaled columns; (**c**) schematic model of the repair/rehabilitated column.

**Figure 3 materials-13-05778-f003:**
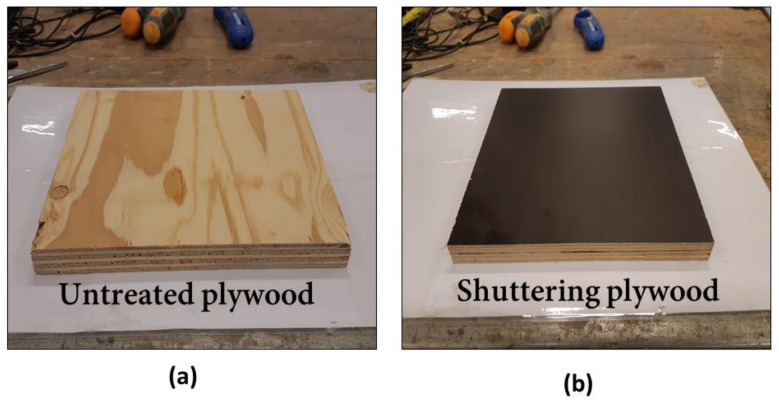
Types of used plywood formwork for composite column—untreated treated plywood (UTT) and shuttering plywood (STT).

**Figure 4 materials-13-05778-f004:**
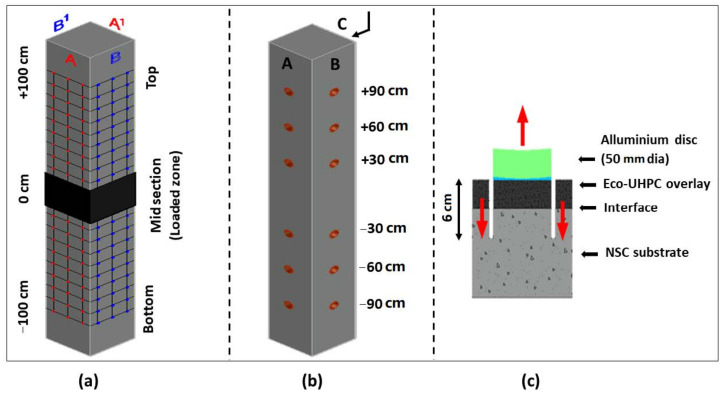
(**a**) Ultrasonic Pulse Velocity (UPV) measuring nodes—Red (A-A’), Blue (B-B’); (**b**) spots for pull-off test (marked in red); (**c**) schematic representation of pull-off test.

**Figure 5 materials-13-05778-f005:**
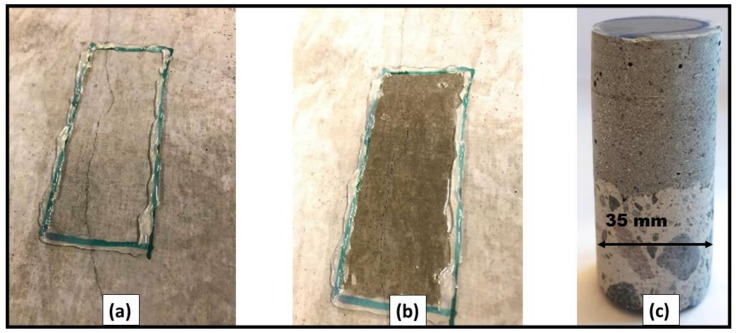
(**a**) Before placing epoxy resin, (**b**) after 24 h set of epoxy resin; (**c**) extracted core going through resin.

**Figure 6 materials-13-05778-f006:**
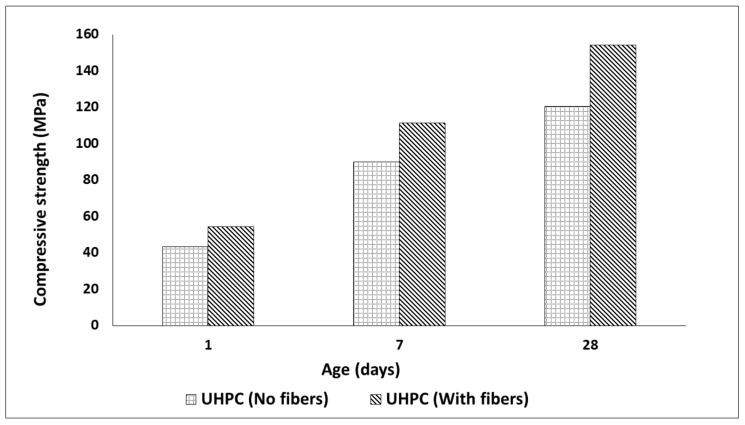
Preliminary UHPC compressive strength development without and with fibers.

**Figure 7 materials-13-05778-f007:**
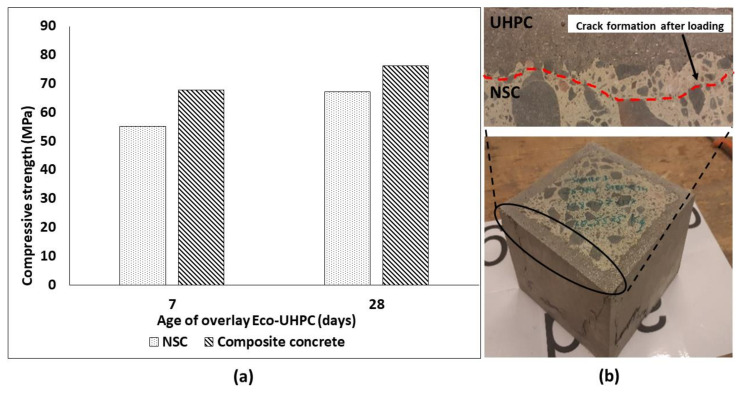
(**a**) Compressive strength development of composite cubes; (**b**) cube failure/crack formation after loading.

**Figure 8 materials-13-05778-f008:**
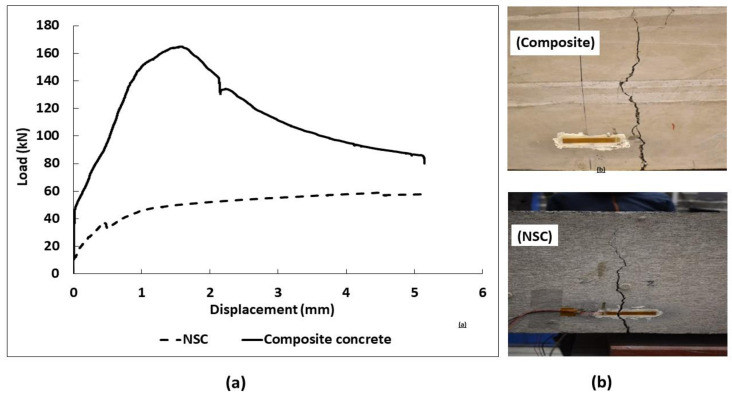
Flexural strength of composite specimen (**a**) load vs. displacement; (**b**) crack development sequence.

**Figure 9 materials-13-05778-f009:**
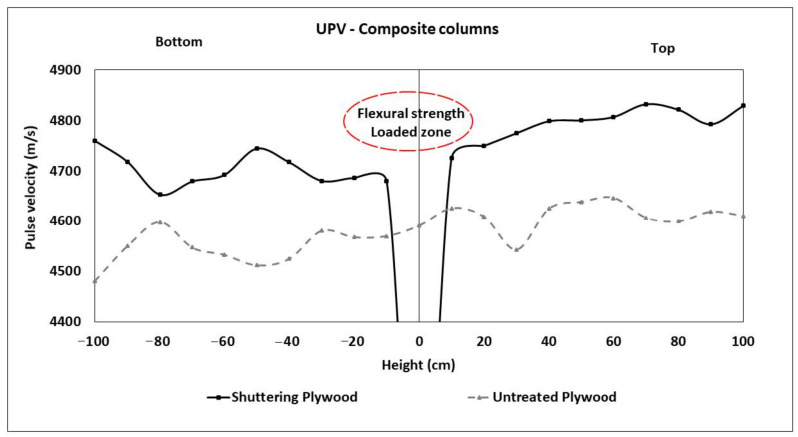
Averaged Ultrasonic Pulse Velocity (UPV) measurement of composite specimens using shuttering and untreated plywood.

**Figure 10 materials-13-05778-f010:**
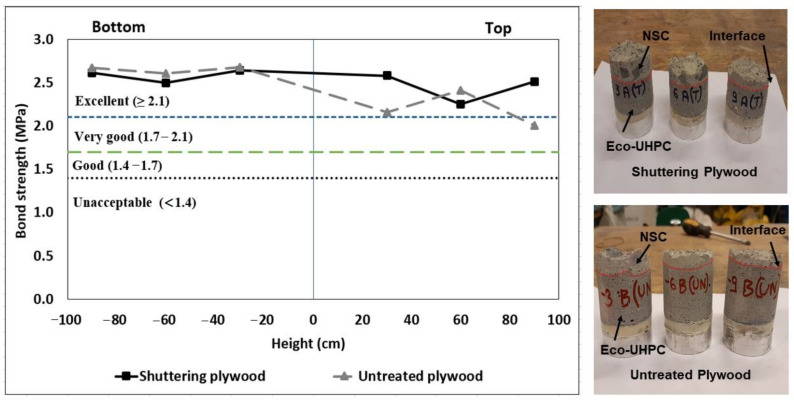
Pull-off test—bond strength development of composite specimens at different sections.

**Figure 11 materials-13-05778-f011:**
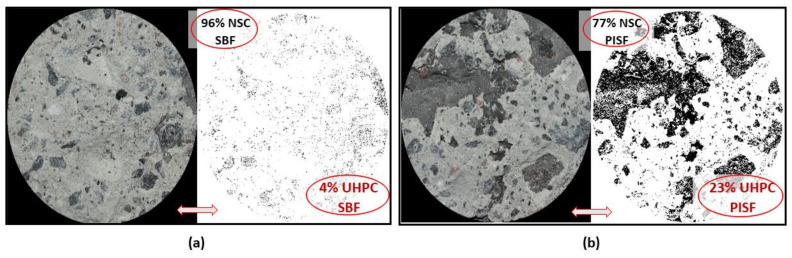
Binarized images of core surface showing fragments of UHPC and NSC (**a**) shuttering plywood; (**b**) untreated plywood.

**Figure 12 materials-13-05778-f012:**
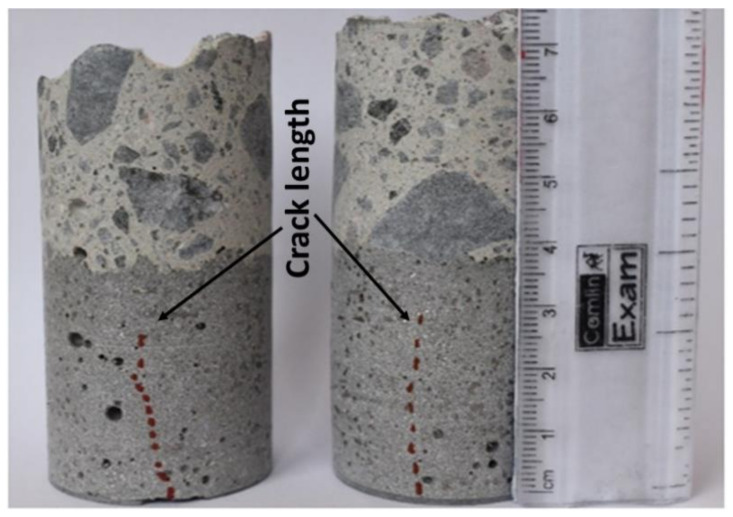
Shrinkage crack—depth measurement.

**Figure 13 materials-13-05778-f013:**
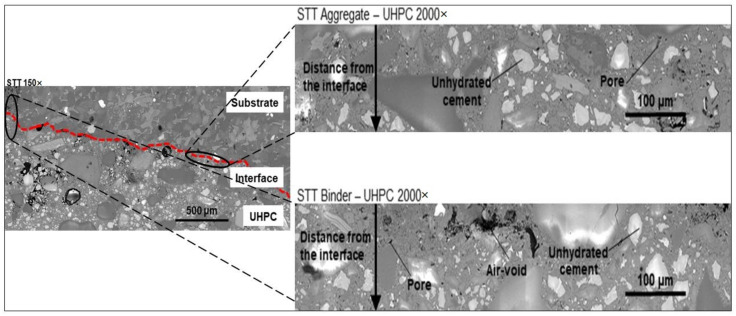
Interfacial bond zone of composite specimen using shuttering plywood at 150× and 2000×.

**Figure 14 materials-13-05778-f014:**
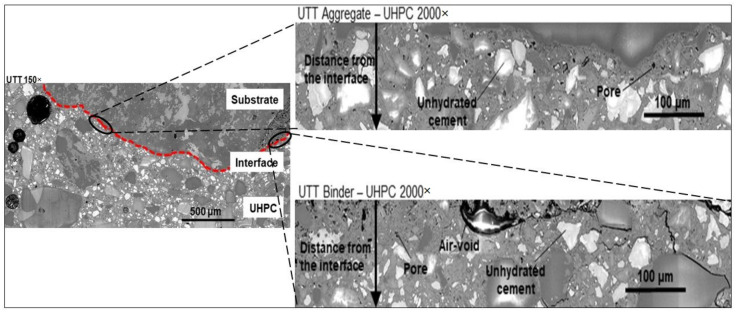
Interfacial bond zone of composite specimen using untreated plywood at 150× and 2000×.

**Figure 15 materials-13-05778-f015:**
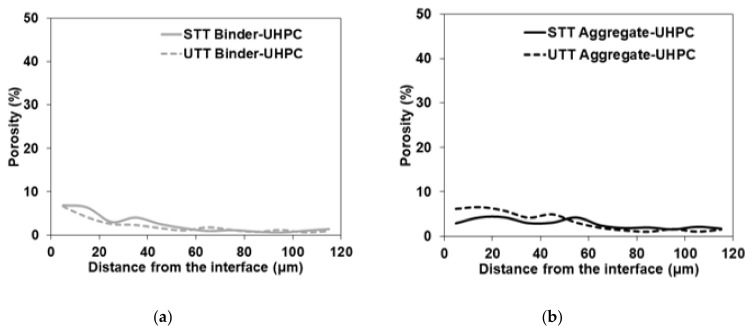
Fraction of porosity away from binder-UHPC and aggregate-UHPC interface for (**a**) shuttering plywood (STT); (**b**) untreated plywood (UTT) formwork.

**Figure 16 materials-13-05778-f016:**
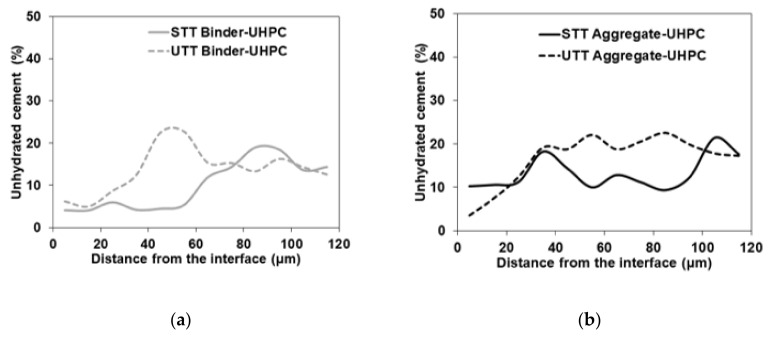
Fraction of unhydrated cement away from binder-UHPC and aggregate-UHPC interface for (**a**) shuttering plywood (STT); (**b**) untreated plywood (UTT) formwork.

**Figure 17 materials-13-05778-f017:**
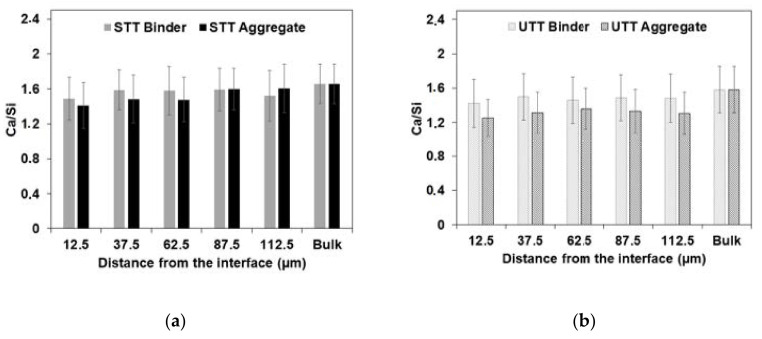
Ca/Si atomic ratio at various distances from binder-UHPC to aggregate-UHPC interfaces of columns using (**a**) shuttering plywood (STT); (**b**) untreated plywood (UTT) formwork.

**Figure 18 materials-13-05778-f018:**
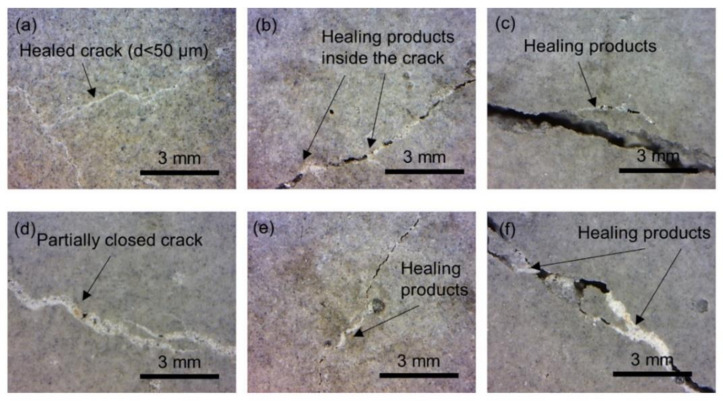
The images of surface cracks on UHPC layer filled with self-healing products (**a**) fully healed crack; (**b**-**f**) partial healed cracks.

**Table 1 materials-13-05778-t001:** Mix proportion and mechanical properties—Eco-UHPC and Normal concrete.

Mix	Materials	Density (kg/m^3^)	Proportion (kg/m^3^)	Percentage Volume
**Eco-UHPC**	Portland Cement (CEM I 42.5N)	3200	651	20.3
	Silica Fume 920D	2000	130.2	6.5
	Limestone	2600	651	25
	Quartz	2650	65.1	2.5
	Sand—B15	2670	227.9	8.5
	Sand—B35	2670	227.9	8.5
	PCE—Superplasticizer (solid + water)	1064 (kg/L)	32.6	3.2
	Steel fibers 6 mm	7500	65.1	0.9
	Steel fibers 13 mm		97.7	1.3
	Air	-	-	4.0
	*w*/*c*—0.33	1000	192	19.2
	**Test age (days)**	**Compressive strength (MPa)**		
	1	54.29		
	7	111.33		
	28	154.25		
**NSC**	**Materials**	**Proportion (kg/m^3^)**		
	Portland Cement (CEM II/A-V 52.5N)	340		
	Dolomit filler—KM200	160		
	Fine aggregate (0–4)	1021		
	Coarse aggregate (8–16)	802		
	MasterGlenium SKY 823	3.4		
	*w*/*c*—0.55	187		
	**Test age (days)**	**Compressive strength (MPa)**		
	1	19.52		
	7	27.00		
	28	53.17		

**Table 2 materials-13-05778-t002:** Chemical composition of the used dry materials.

Chemical (%)	Cement I 42.5 N	Silica Fume	Quartz	Sand (B15, B35)
CaO	63.30	1	99.6	-
SiO_2_	21.20	≥85	-	90.5
Al_2_O_3_	3.40	1	0.25	4.9
Fe_2_O_3_	4.12	1	0.02	0.5
MgO	2.20	1	-	-
Na_2_O	0.18	0.5	-	1.2
K_2_O	0.56	1.2	-	2
SO_3_	2.70	2	-	-
Cl	<0.01	0.3	-	-
LOI	2.50	4	0.15	-

**Table 3 materials-13-05778-t003:** Characteristic of used steel fibers.

Properties	DM 6/0.175	DG 13/0.3 − E430
Material	Steel − brass coated	Stainless steel
Type of fiber	Wire fiber − microfiber	Wire fiber − straight steel
Length (L)	6 mm	13 mm
Diameter (d)	0.175 mm	0.3 mm
Ratio (L/d)	34.3	43
Tensile strength (MPa)	2800	1100
Modulus of elasticity (GPa)	210	200
Quantity of fibers/kg	882,000	144,174

**Table 4 materials-13-05778-t004:** Pull-off test bond failure modes.

ID	Bond Failure Mode
SBF	Substrate failure
OF	Overlay failure
IF	Interface failure
PISF	Partial interface-substrate failure

**Table 5 materials-13-05778-t005:** Pull-off test failure modes of composite specimens using shuttering and untreated plywood for casting. Bond failure mode ID—SBF is substrate failure; OF is overlay failure; IF is interface failure; and PISF is partial interface–substrate failure.

		Shuttering Plywood (STT)			Untreated Plywood (UTT)		
		Side			Side		
Section	Level	A	B	C	A	B	C
**Top**	+90	SBF	SBF	SBF	PISF	PISF	PISF
	+60	SBF	SBF	SBF	SBF	SBF	PISF
	+30	SBF	SBF	SBF	PISF	SBF	PISF
**Bottom**	−30	SBF	PISF	SBF	SBF	SBF	SBF
	−60	SBF	SBF	SBF	SBF	SBF	SBF
	−90	SBF	SBF	SBF	SBF	SBF	SBF

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
