# Peer review of "Eco-UHPC as Repair Material—Bond Strength, Interfacial Transition Zone and Effects of Formwork Type"

_materials, 2020, doi:10.3390/ma13245778_

Round 1
Reviewer 1 Report
1)This paper reports experimental study of UHPC as a repair material. Some good results are obtained. Overall it is clearly presented.
2) Why did the authors use two types of cement for UHPC and NSC? And please give the difference between them.
3) UHPC was used as a thin over layer with thickness of 15 or 30 mm. What about the distribution and orientation of steel fibers in this material.
4) What useful information can be observed from Fig. 1. It is better to give SEM photos with the same magnification.
Reviewer 2 Report
Taking into account the objectives and scope of the Journal, the document has an appropriate scope and level. The topic is original and interesting. Anyway, the document can be improved for a better understanding
What is the full-scale applicability of high temperature curing?
The incoporation of limestone can favor carbonation process and induce the depassivation and subsequent oxidation of the steel fibers. What can the authors comment about this?
Point 3 should be divided into sub-sections because it is confusing and difficult to correctly understand the discussion of the results.
The conclusions seem to me to be correct and well formulated. I miss some conclusion about the applicability of the research and its possible uses in the field of engineering and construction.
This first version of paper can be considered positively with minor changes.
Reviewer 3 Report
- Please present in abstract: globalism of the issue, and the main results of the research.
- Quoting: „The properties of the formed ITZ depend on several factors including, e.g. substrate roughness, moisture content and w/c ratio [10–12].” At this point, a multi-criteria technical-economic analysis should be carried out, but it is on the occasion of the next article, e.g. taking into account, among other things, the placement rate and UHPC pressure.
- Quoting: „The bond strength was higher for rough and grooved substrates, [14,15].” In the 1980s, the Japanese built several buildings in monolithic technology, where the concrete mixture of about 0.25 w/c with the addition of microsilica was fed in a grooved transparent plastic formwork. This formwork increased the compressive strength of the concrete by about 50%. Now there is a quality leap and we are dealing with UHPC. Here are laboratory tests and repair methods. Has there been any simulation of the implementation of this solution in industry?
- Quoting: „This cement consumption consequently emit 2-3 times more CO2 and thus significantly higher carbon footprint [20][21].” „Hence, its partially replacement with for example limestone filler will not only reduce the CO2 emission, but will also lower the production cost, lower the shrinkage and reduce the crack risk, [25–30].” Limestone must also be produced (CO2 emission, economics). Hence this multi-criteria analysis.
- Figure 1. Maybe to put this article in chronological order. Here we have the technological issues, the functional properties, microstructural properties and so on. The description of the apparatus for fig.1 can be found only at fig.6. Here are sprayed specimens, further polished, another SEM technique.
- Please specify the technology of laying the UHPC concrete mix (feed rate, pressures, concrete temperature). And also the technology of dismantling the work forms. Are other technologies foreseen in the industry?
- Quoting: „Also the used limestone powder could promote formation of carboaluminate and C-S-H, [61].” Are you sure? We rely only on this literature. In future, please also use XRD research. The C-S-H phase is always amorphous, but with this amount of limestone in the UHPC (25%), C-S-H, depending on the temperature and feed rate of the concrete mix, can be crystallized and perhaps it goes towards the jenite. But this is only visible on XRD.
- Please delete the first conclusion. That is obvious.
- Please come up with more precise conclusions, in the content of the article they are.
Reviewer 4 Report
In the introduction almost all necessary information are presented and the introduction is well organized. Presented literature is limited to the newest knowledge and is mainly from 2000 till 2020. The methodology is sufficiently explained that someone else, who has the equipment and required knowledge about the topic, could repeat the study. The conclusions are presented clearly and compared with other studies. However there are some remarks which would be beneficial in order to improve the manuscript:
-Despite the fact that the presented literature is limited to the newest knowledge and is mainly from 2000 till 2020, still there is a lack of some references improtant in this topic such as:
Adhesion in Layered Cement Composites, ISBN 978-3-030-03783-3,
Near-to-surface properties affecting bond strength in concrete repair, https://doi.org/10.1016/j.cemconcomp.2013.11.005
-There is slight inconsistency between the diameter of the sample tested using pull-off method and cores drilled out of the samples.
-Also there is another inconsistency which is seen in the picture 12. Because this research are describing the bonding in the concrete repair it is important that adhesion between the layers should be not less than 1.5 MPa. Thus Reveiwer cannot agree that adhesion values bellow 0.7-1.4 are treated as "fair" because they are unacceptable.
Overall Merit of the Manuscript in Reviewer's opinion is very high and because the author's obtained values of adhesion above 2.0 MPa made these research acceptable in accordance to the literature.
Also the Manuscript is within the scope of the journal and can be accepted after some minor revision.
